# A data-driven eXtreme gradient boosting machine learning model to predict COVID-19 transmission with meteorological drivers

**Md. Siddikur Rahman** 📧 *, **Arman Hossain Chowdhury** 📧

Department of Statistics, Begum Rokeya University, Rangpur, Rangpur, Bangladesh

* siddikur@brur.ac.bd

**Data Availability Statement:** All relevant data are within the paper and its Supporting information files.

**Funding:** The author(s) received no specific funding for this work.

## Abstract

COVID-19 pandemic has become a global major public health concern. Examining the meteorological risk factors and accurately predicting the incidence of the COVID-19 pandemic is an extremely important challenge. Therefore, in this study, we analyzed the relationship between meteorological factors and COVID-19 transmission in SAARC countries. We also compared the predictive accuracy of Autoregressive Integrated Moving Average (ARIMAX) and eXtreme Gradient Boosting (XGBoost) methods for precise modelling of COVID-19 incidence. We compiled a daily dataset including confirmed COVID-19 case counts, minimum and maximum temperature (˚C), relative humidity (%), surface pressure (kPa), precipitation (mm/day) and maximum wind speed (m/s) from the onset of the disease to January 29, 2022, in each country. The data were divided into training and test sets. The training data were used to fit ARIMAX model for examining significant meteorological risk factors. All significant factors were then used as covariates in ARIMAX and XGBoost models to predict the COVID-19 confirmed cases. We found that maximum temperature had a positive impact on the COVID-19 transmission in Afghanistan (β = 11.91, 95% CI: 4.77, 19.05) and India (β = 0.18, 95% CI: 0.01, 0.35). Surface pressure had a positive influence in Pakistan (β = 25.77, 95% CI: 7.85, 43.69) and Sri Lanka (β = 411.63, 95% CI: 49.04, 774.23). We also found that the XGBoost model can help improve prediction of COVID-19 cases in SAARC countries over the ARIMAX model. The study findings will help the scientific communities and policymakers to establish a more accurate early warning system to control the spread of the pandemic.

## Introduction

The novel coronavirus disease 2019 (COVID-19), induced by severe acute respiratory syndrome 2 [1, 2], has become a serious public health threat globally. The disease has quickly spread over the world because of its extremely human-to-human transmission characteristics [3–5]. As of July 02, 2022, more than 553.87 million confirmed cases and over 6.36 million deaths have been reported globally [6]. It has already been studied that meteorological factors like temperature, relative humidity and wind speed have been linked to the development of the

**Competing interests:** The authors have declared that no competing interests exist.

transmission of recognized coronavirus infections such as Severe Acute Respiratory Syndrome (SARS) and Middle East Respiratory Syndrome Coronavirus (MERS-CoV) [7, 8]. According to laboratory tests, SARS-CoV-2 is very stable in cold conditions but vulnerable to rising temperatures [9]. Different previous studies also investigated that meteorological factors such as temperature [10–12], humidity [13, 14], and wind speed [15] might affect COVID-19 transmission [16–18]. The transmission of the COVID-19 pandemic is reduced as temperature rises in China as well as other regions of the world [13, 16, 18]. It was also found that wind speed had lagged correlations with COVID-19 incidence in various Turkish cities [19]. Humidity was also a major meteorological factor in reducing COVID-19 viral transmission in China, Pakistan, Sri Lanka and other countries [11, 14, 20]. However, the humidity was also negatively associated with the COVID-19 epidemic in Indonesia and New York [21, 22].

Different studies widely used different types of statistical approaches including correlation, regression analysis, generalized additive model, and generalized linear model to analyze the influence of environmental variables on COVID-19 transmission [5, 13, 19, 21–24]. Besides these, several studies have used Autoregressive Integrated Moving Average with exogeneous variables model to determine the association of climate variables with COVID-19 transmission and forecasting [23, 25, 26]. Time-series modelling is a popular forecasting method for understanding the dynamic association of important variables. However, the transmission of COVID-19 disease is often influenced by several factors which exhibit nonlinear influences which cause problems [27]. This problem can be easily solved by machine learning techniques [28, 29]. Given the uncertainty around decisions on the accurate time of the emergence and disappearance of the disease, short-term forecasting is crucial to create better plans and more appropriate responses. The eXtreme Gradient Boosting (XGBoost) is an uptrend machine learning technique in time series modelling. The XGBoost model can generate a high precision result for its self-learning characteristics. This study contributes to the advancement of the time-series prediction of COVID-19. Consequently, an initial benchmarking is given to demonstrate the potential of machine learning for future research. The study further suggests that a genuine novelty in COVID-19 prediction can be realized by a data-driven XGBoost machine learning model. Currently, no study used this technique for determining the association between meteorological factors and COVID-19 transmission and prediction. Therefore, the study aimed to: (a) identify the meteorological risk factors; (b) compare the predictive accuracy of the ARIMAX and XGBoost for precise modelling of COVID-19 incidence in the South Asian Association for Regional Cooperation (SAARC) countries.

In this study, our proposed methodology (Fig 1) and results are useful to select a suitable model for COVID-19 prediction. The findings from this study will help the countries' policymakers for taking effective strategies to establish a more accurate early warning system to control the spread of the pandemic.

## Materials and methods

### Data source

The daily COVID-19 confirmed cases data of the SAARC countries (Afghanistan, Bangladesh, Bhutan, India, Maldives, Nepal, Pakistan, and Sri Lanka) were collected from the Johns Hopkins Coronavirus Resource Center [30]. The meteorological data of each country were obtained based on hourly meteorological observations from the NASA Langley Research Center (LaRC) website [31], including minimum and maximum temperatures (˚C), relative humidity (%), maximum wind speed (m/s), surface pressure (kPa) and precipitation (mm/day). The study period was from the onset of COVID-19 to January 29, 2022, for each SAARC country.

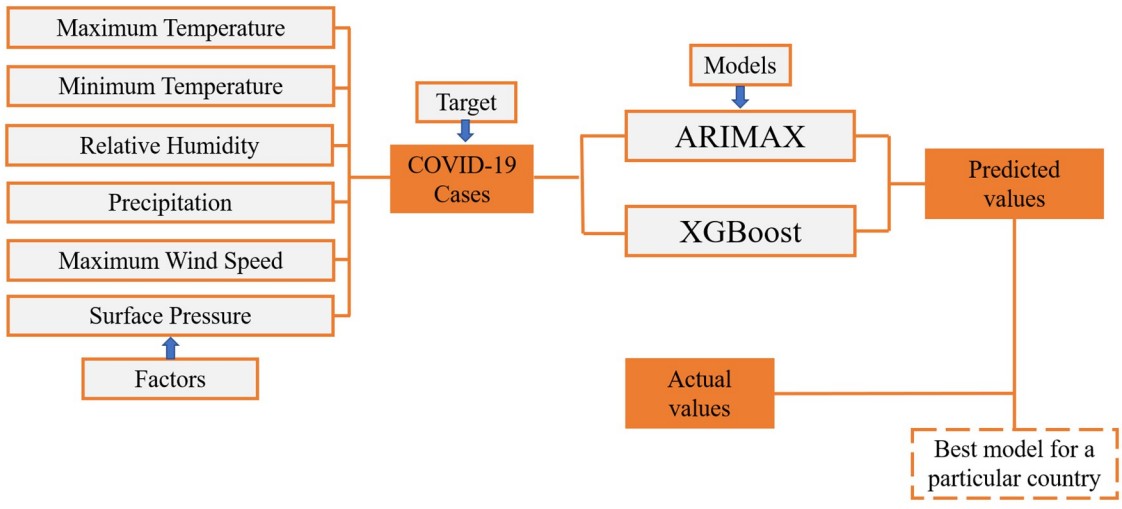

**Fig 1. Schematic of the proposed methodology.**

## Model building, prediction, and performance evaluation

Predictive modeling and statistical analyses were conducted using RStudio (Version 4.1.0) [32]. The 'tseries' and stats packages were used to process the time series. The ARIMAX models were built with the 'forecast' package using auto.arima function for choosing the best model based on the Corrected Akaikes Information Criteria (AICc) values [33]. The 'forecastxgb' package was used for building the XGBoost model. Details data and necessary codes for predictive modeling and statistical analysis are provided in supplements (S1 Table and S1 Text).

In this study, predictive accuracy of ARIMAX and XGBoost models was compared to determine which was more suitable for predicting COVID-19 confirmed cases in SAARC countries based on meteorological risk factors. The data were divided into training and test sets. All the significant meteorological factors were used as covariates in ARIMAX and XGBoost models for predicting the COVID-19 confirmed cases. The ensemble machine learning technique XGBoost were built using the lagged meteorological variables as covariates by frequently changing several parameters. The adjusted parameters for the model of each SAARC country are nrounds, nrounds_method = 'cv', nfold, seas_method, trend_method = 'none'.

Predictive accuracy refers to the capacity of the model to predict COVID-19 incidence. There are several metrics for computing the model's accuracy [34]. However, in this study, we used four prominent performance metrics such as the mean absolute percentage error (MAPE), mean percentage error (MPE), mean absolute error (MAE) and root mean square error (RMSE). The mathematical form of the error measures are as follows:

$$MAPE = \frac{1}{n}\sum\nolimits_{i=1}^{n}\left|\frac{\hat{y}_i - y_i}{y_i}\right| \times 100\% \tag{1}$$

$$MAE = \frac{1}{n}\sum\nolimits_{i=1}^{n}|\hat{y}_i - y_i| \tag{2}$$

$$RMSE = \sqrt{\frac{1}{n}\sum\nolimits_{i=1}^{n}(\hat{y}_i - y_i)^2} \tag{3}$$

$$MPE = \frac{1}{n}\sum_{i=1}^{n}\left(\frac{\hat{y}_i - y_i}{y_i}\right) \times 100\% \tag{4}$$

Where n represents the number of observations, $\hat{y}_i - y_i$ represents the error between the predicted and actual value.

## ARIMAX model

The Autoregressive Integrated Moving Average (ARIMA) model introduced by Box and Jenkins (2013) is widely used for predicting time series data because of its capacity to handle non-stationary data [35]. ARIMA(p, d, q) combines the Autoregressive (AR) and Moving Average (MA) models, with the 'I' indicating integration; where p stands for autoregressive order, d for differencing order, and q stands for moving average order [36]. The AR(p) in ARIMA stands for a linear combination of p prior observations with a random error factor that determines a variable's future value which can be mathematically expressed as

$$Y_t = C + \emptyset_1 Y_{t-1} + \emptyset_2\ Y_{t-2} + \emptyset_3\ Y_{t-3} + \emptyset_4\ Y_{t-4}\ldots..\emptyset_p Y_{t-p} + \varepsilon_t \tag{5}$$

Where, $Y_t$ and $\varepsilon_t$ are the actual value and random error terms at time t, $\emptyset_i$ (i = 1,2,3,4. . ..) represents model parameters, and c is a constant. The order of the model is a positive integer p. The MA(q) model incorporates a dependent variable for previous errors which can be expressed as

$$Y_t = \mu + \theta_1\varepsilon_{t-1} + \theta_2\varepsilon_{t-2} + \theta_3\varepsilon_{t-3} + \theta_4\varepsilon_{t-4} + \cdots + \theta_q\varepsilon_{t-q} + \varepsilon_t \tag{6}$$

Where $\mu$ indicates the series mean, $\theta_j$ (j = 1, 2, 3 . . . q) indicates model parameters, and q is the model's order [37].

The ARIMA model may be stated in its basic form as

$$y_t' = c + \emptyset_1 y_{t-1}' + \emptyset_2 y_{t-2}' + \ldots + \emptyset_p y_{t-p}' + \theta_1\varepsilon_{t-1} + \theta_2\varepsilon_{t-2} + \ldots + \theta_q\varepsilon_{t-q} + \varepsilon_t \tag{7}$$

where $y_t'$ represents differenced series (it can be more than one); $\emptyset_1, \emptyset_2, \ldots \emptyset_p$ are the coefficients of AR terms and $\theta_1, \theta_2 \ldots \theta_q$ are the coefficients of moving average term.

The ARIMAX model is the generalization of the ARIMA model. It enhances the ARIMA model's capabilities by including several meteorological information such as temperature, humidity, precipitation and other meteorological conditions in time series modelling. An ARIMAX model is be formed as follows:

$$y_t = \beta_0 + \beta_1 x_{1,t} + \cdots + \beta_k x_{k,t} + \eta_t \tag{8}$$

where, $y_t$ represents the response variable for the given time series; $x_{1,t} \ldots x_{k,t}$ are the features or exogenous variables of the time series that potentially explain $y_t$; $\eta_t$ is the regression model error that describes the ARIMA model (Eq 3) [38].

## XGBoost model

The XGBoost model is a supervised machine learning technique and an emerging machine learning method for time series forecasting in recent years [39, 40]. It uses an improved generalized gradient boosting library that can rapidly assess the value of all input attributes [41–43]. Boosting is a technique that combines hundreds of low-accuracy prediction models into a single high-accuracy model by frequently integrating the models under tolerable parameter values

[44–46]. The objective function of the model is as follows:

$$Obj^{(t)} = \sum_{i=1}^{n} l\left(y_i, \hat{y}_i^{(t-1)} + f_t(x_i)\right) + \Omega(f_t) + constant \tag{9}$$

Where $y_i$ stands for the observed values, $x_i$ stands for the feature vector, n stands for the sample size, m stands for the number of iterations, and $f_m$ stands for the error in m iterations. $l$ stands for the loss function, which computes the deviation between the label and the forecasting in the previous phase as well as the output of the new tree, and R stands for the regularization term, which reduces the new tree's output variation [37, 39, 47].

## Result

As of 29 January 2022, India had reported the highest total of 41.1 million COVID-19 confirmed cases, resulting in 0.5 million fatalities, whereas Bhutan had reported the lowest COVID-19 confirmed cases and fatalities among SARRC countries (Table 1).

The maximum temperature among the SAARC countries varies from -3.38˚C (Nepal) to 47.01˚C (India) and the minimum temperature varies from -26.17˚C (Afghanistan) to 31.67˚C (India). The highest average maximum temperature was observed in India (32.54˚C). The highest level of humidity was observed in Maldives (100%), but the lowest level of humidity was observed in Afghanistan (5.06%). Bangladesh had the highest maximum wind speed at 10M (15.68 m/s), but Bhutan had the lowest (1.62 m/s). The highest surface pressure was observed in Bangladesh (101.88 kPa) and the lowest surface pressure was observed in Nepal (66.08 kPa) as illustrated in Fig 2.

The time series figure depicts the trend of COVID-19 confirmed cases from the onset of the disease to January 29, 2022, in each SAARC country. Daily confirmed cases in Bangladesh, Nepal and Pakistan fluctuated at different periods including a highly upward trend. The pattern in Afghanistan and Sri Lanka was remarkably similar, indicating a downward tendency. Overall, Bhutan and Maldives had a comparatively lower rate of COVID-19 transmission than other SAARC countries (Fig 3). The cross-correlation between COVID-19 confirmed cases and meteorological variables was formed at 0 to 30 lags. Only positive lags were considered to explore the influence of meteorological factors on the COVID-19 transmission in a certain period [48]. In Afghanistan, the maximum and minimum temperature at lag 0 showed a significant relationship with COVID-19 confirmed cases. The only maximum temperature at lag 4 showed a significant relationship in India. Maximum wind speed showed a significant relationship in Bangladesh at lag 9 and Maldives at lag 13 days. Relative humidity at a lag of 26

**Table 1. Summary statistics of COVID-19 confirmed cases and deaths for SAARC countries till January 29, 2022.**

| Countries | Daily confirmed cases | | | | Daily Deaths | | | |
|---|---|---|---|---|---|---|---|---|
| | Min | Max | Mean ± SD | Total | Min | Max | Mean ± SD | Total |
| Afghanistan | 0 | 3243 | 228.50 ± 397.09 | 161,306 | 0 | 159 | 10.49 ± 18.95 | 7405 |
| Bangladesh | 0 | 16,230 | 2559 ± 3163.93 | 1,773,149 | 0 | 264 | 40.88 ± 53.22 | 28,329 |
| Bhutan | 0 | 205 | 6.57 ± 19.39 | 4566 | 0 | 1 | 0.005 ± 0.08 | 4 |
| India | 0 | 533,035 | 56,214 ± 85,201.04 | 41,092,522 | 0 | 4529 | 665.80 ± 904.03 | 486,718 |
| Maldives | 0 | 2813 | 192.30 ± 390.21 | 133,288 | 0 | 10 | 0.40 ± 0.98 | 274 |
| Nepal | 0 | 10,052 | 1287.20 ± 1937.83 | 947,394 | 0 | 619 | 15.90 ± 39.42 | 11,703 |
| Pakistan | 0 | 12,073 | 2011 ± 1758.87 | 1,417,991 | 0 | 313 | 41.49 ± 38.87 | 29,248 |
| Sri Lanka | 0 | 11,366 | 829.80 ± 1237.04 | 609,047 | 0 | 334 | 20.98 ± 42.64 | 15,400 |

Min: Minimum; Max: Maximum; SD: Standard Deviation.

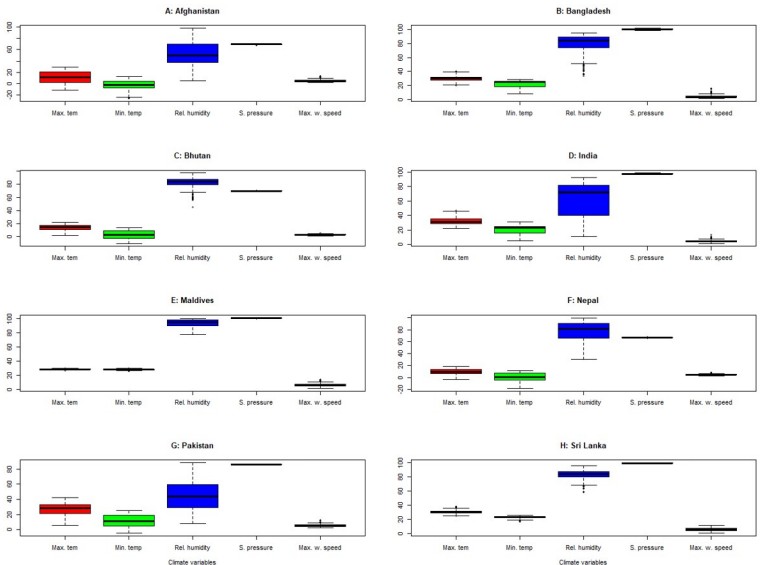

**Fig 2. Boxplot of meteorological variables for SAARC countries.** Max. tem: Maximum temperature; Min. temp: Minimum temperature; Rel. humidity: Relative humidity; S. pressure: Surface pressure; Max. w. speed: Maximum wind speed.

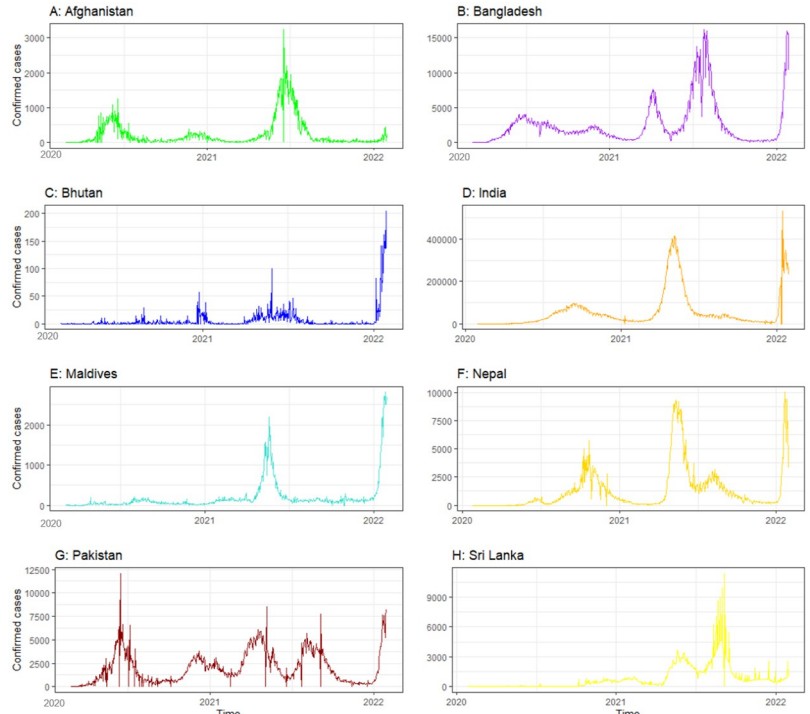

**Fig 3. Time series plot showing the trend of COVID-19 confirmed cases for SAARC countries.**

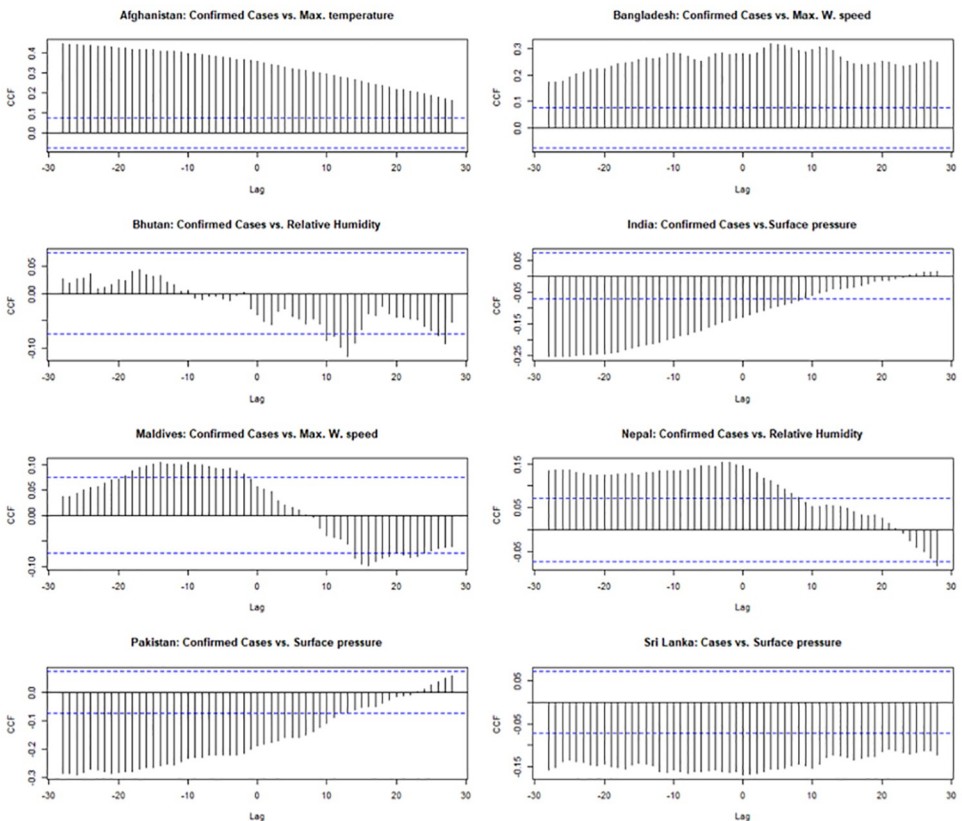

**Fig 4. Cross-correlation between COVID-19 confirmed cases and meteorological variables in SAARC countries.**
Max. temperature: Maximum temperature; Max. W. speed: Maximum Wind speed; CCF: Cross-Correlation Function.

days in Bhutan and lag of 10 days in Nepal showed a significant correlation with COVID-19 confirmed cases. Surface pressure showed a significant correlation with COVID-19 confirmed cases in India at lag of 9 days, in Sri Lanka at lag of 13 days and in Pakistan at lag of 28 days (Fig 4).

The aforementioned meteorological factors were used as covariates in ARIMAX model at different lags to determine their influence on COVID-19 confirmed cases. For example, in Afghanistan, the maximum and minimum temperature at lag 0 was used as covariates for building the ARIMAX model. Similarly for Bangladesh, Bhutan, India, Maldives, Nepal, Pakistan and Sri Lanka, the lagged variables were used as covariates and the influence of those variables on the disease was shown in Table 2.

Table 2 depicts the minimum temperature with a lag of 0 (i.e., same day) in Afghanistan ($\beta$ = -8.93, 95% CI: -14.30, -3.56) negatively impact the transmission of COVID-19 cases. The maximum temperature with a lag of 4 days in India ($\beta$ = 0.18, 95% CI: 0.01, 0.35) and with a lag of 0 (i.e., same day) in Afghanistan ($\beta$ = 11.91, 95% CI: 4.77, 19.05) had a positive influence on the transmission of COVID-19 confirmed cases. Maximum wind speed with a lag of 9 days in Bangladesh ($\beta$ = -53.89, 95% CI: -93.45, -14.32) and a lag of 13 days in Maldives ($\beta$ = -4.24, 95% CI: -8.31, -0.18) negatively impacts the transmission of COVID-19 confirmed cases. Relative humidity with a lag of 10 days in Nepal ($\beta$ = -4.84, 95% CI: -9.20, -0.48) and at a lag of 26 days in Bhutan ($\beta$ = -0.12, 95% CI: -0.22, -0.02) negatively impacts COVID-19 confirmed cases. Surface pressure positively impacts COVID-19 confirmed cases in Pakistan ($\beta$ = 25.77,

**Table 2. Estimated parameters with 95% confidence intervals of significant meteorological factors of ARIMAX models.**

| Factors | Afghanistan | Bangladesh | Bhutan | India | Maldives | Nepal | Pakistan | Sri Lanka |
|---|---|---|---|---|---|---|---|---|
| | ARIMAX (3,1,0) | ARIMAX (0,1,0) | ARIMAX (5,1,0) | ARIMAX (2,1,0) | ARIMAX (1,1,0) | ARIMAX (1,1,0) | ARIMAX (4,1,0) | ARIMAX (5,1,0) |
| Min. temperature (0) | -8.93* (-14.30, -3.56) | | | | | | | |
| Max. temperature (0) | 11.91* (4.77, 19.05) | | | | | | | |
| Max. temperature (4) | | | | 0.18* (0.01, 0.35) | | | | |
| Max. W. speed (9) | | -53.89* (-93.45, -14.32) | | | | | | |
| Max. W. speed (13) | | | | | -4.24* (-8.31, -0.18) | | | |
| Rel. humidity (10) | | | | | | -4.84* (-9.20, -0.48) | | |
| Rel. humidity (26) | | | -0.12* (-0.22, -0.02) | | | | | |
| Surface pressure (9) | | | | -1.91* (-3.75, -0.06) | | | | |
| Surface pressure (13) | | | | | | | | 411.63* (49.04, 774.23) |
| Surface pressure (28) | | | | | | | 25.77* (7.85, 43.69) | |

Max. temperature: Maximum temperature; Min. temperature; Minimum temperature; Rel. humidity: Relative humidity; Max. W. speed: Maximum Wind speed;

* indicates significance at 5% level.

95% CI: 7.85, 43.69) with a lag of 28 days and Sri Lanka (β = 411.63, 95% CI: 49.04, 774.23) with a lag of 13 days. Moreover, surface pressure with a lag of 9 days in India (β = -1.91, 95% CI: -3.75, -0.06) negatively impacts the transmission of COVID-19 confirmed cases. The detailed result about the influence of meteorological factors on COVID-19 transmission is presented in Table 2.

The average value of the error measures in the XGBoost model is lower than the ARIMAX models for all the SAARC countries (Fig 5). Hence, in our study, it was found that XGBoost performs better in predicting COVID-19 confirmed cases in most of the SAARC countries. The detailed procedure of ARIMAX and XGBoost model fitting for COVID-19 confirmed case prediction is presented in S1 File.

## Discussion

This study predicted the effect of meteorological factors on the transmission of COVID-19 confirmed cases in SAARC countries. In South Asia, Bangladesh experiences subtropical monsoon weather, with annual average temperatures hovering from 26 to 36˚C [49]. Afghanistan experiences hot, dry summer and chilly winter. In summer, the highest temperature in the country reaches up to 50˚C, whereas in winter it is -25˚C [50]. India has two distinct climate conditions: tropical monsoon weather and tropical wet and dry weather [51]. In Pakistan, there has a wide range of typical temperatures, from 2˚C to 38˚C [52]. There are distinct rainy and dry seasons in Sri Lanka's tropical climate. The coastal regions of Sri Lanka get year-round temperatures of 28˚C whereas the highland regions experience lower, more moderate

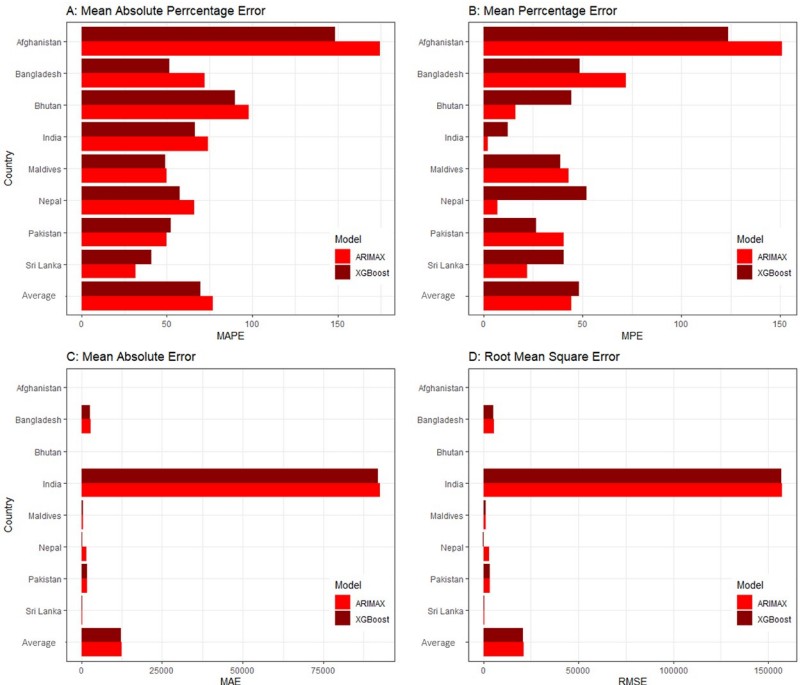

**Fig 5. Performance metrics of the ARIMAX and XGBoost models for predicting COVID-19 confirmed cases in SAARC countries.**

temperatures of 16˚C to 20˚C [53]. The vast elevational differences in Bhutan result in a diversified climate [54]. The climate of Nepal varies according to altitude: subtropical with a rainy season in the southern flat strip, moderate in the low mountains, and chilly in the Himalayan peaks [55].

The study found that the meteorological factors have both positive and negative influences on the transmission of COVID-19 confirmed cases. For instance, the maximum temperature had a positive influence on the transmission of COVID-19 confirmed cases in Afghanistan and India which is similar to some previous studies in the EU [56]. This study also found a negative impact of minimum temperature on COVID-19 transmission which is in line with some previous studies conducted in China and the USA [10, 16, 57]. But some previous studies conducted in Spain and China claimed that temperature had no impact on COVID-19 transmission [58, 59]. We also found that relative humidity had a negative influence on the transmission of COVID-19 cases in Bhutan and Nepal which is in line with some previous studies [10, 21, 22, 60]. We found that surface pressure had a positive influence in Pakistan and Sri Lanka as well as negative impact on the COVID-19 confirmed cases in India which is also in line with some previous studies [61, 62]. It was also stated by some studies that surface pressure had no impact on COVID-19 transmission [25]. This study also found a statistically significant association of maximum wind speed with COVID-19 confirmed cases in Bangladesh and Maldives which is similar to a previous study [63]. This study didn't find any statistically significant association of precipitation with COVID-19 confirmed cases while previous studies examined that this is associated with the transmission of COVID-19 confirmed cases [64].

This paper evaluated the applicability of two popular models ARIMAX and XGBoost for predicting the COVID-19 incidence in SAARC countries. The models showed promising results in terms of predicting the time series without the assumptions that traditional

epidemiological models require. Machine learning models, as an alternative to epidemiological models, showed potential for COVID-19 prediction. Considering the availability of only a small amount of training data, it is expected that machine learning will be developed further as the basis for, or a component of, future COVID-19 outbreak prediction models. The XGBoost model is an uptrend machine learning technique in time series modelling. The novelty of our study is that we predicted COVID-19 confirmed cases with ARIMAX model and a data-driven eXtreme Gradient Boosting algorithm using the significant meteorological factors as covariates. The XGBoost technique offers several benefits in terms of model forecasting, including the non-requirement of data preprocessing, complete feature extraction and high prediction accuracy. This study used this technique to predict COVID-19 confirmed cases using the significant meteorological variables. Because it features a higher late trimming penalty than a standard Gradient boosting decision tree, which reduces the likelihood of overfitting [65]. The XGBoost model was developed by adjusting its different parameters. We selected the most traditional ARIMAX as a baseline for our study. The study found that the XGBoost model performs better in predicting the COVID-19 confirmed cases in most of the SAARC countries. In this study, we used these models as a case study to find the significant relationship between meteorological factors and the COVID-19 transmission and compared the prediction accuracy of those models to determine the best model. The findings of this study are also useful for all other COVID-19-affected countries similar to SAARC countries.

## Limitations

This study used ARIMAX and XGBoost predictive models to investigate the impact of meteorological factors on COVID-19 transmission in SAARC countries. Therefore, a limitation of the study is that, for example, socioeconomic, demographic, healthcare facilities, human mobilities and population density covariates were not incorporated in this study. These covariates might be correlated with the COVID-19 transmission and should be investigated further based on the data availability.

## Conclusion

This study shows the machine learning-based XGBoost model performs better than the ARIMAX model in predicting the COVID-19 incidence in SAARC countries. In the absence of effective COVID-19 prevention strategies, our proposed predictive model is useful for government authorities, researchers and planners to put forward strategic plans to control the spread of COVID-19. It is, therefore, possible for other nations to adopt the suggested frameworks and prevention measures. By exploring the influence of meteorological risk factors on COVID-19 transmission, we can help people to establish a more accurate early warning system and recommend developing appropriate environmental policies to control the spread of the pandemic.

## Supporting information

**S1 Table. Time series COVID-19 data of SAARC countries along with meteorological variables from the onset of COVID-19 incidence to January 29, 2022.**
(XLSX)

**S1 Text. R codes.**
(TXT)

**S1 File.**
(DOCX)

## Acknowledgments

We are grateful to research student Md. Selim Mondol for his generous support and help.

## Author Contributions

**Conceptualization:** Md. Siddikur Rahman.

**Data curation:** Md. Siddikur Rahman, Arman Hossain Chowdhury.

**Formal analysis:** Md. Siddikur Rahman, Arman Hossain Chowdhury.

**Funding acquisition:** Arman Hossain Chowdhury.

**Investigation:** Md. Siddikur Rahman, Arman Hossain Chowdhury.

**Methodology:** Md. Siddikur Rahman, Arman Hossain Chowdhury.

**Project administration:** Md. Siddikur Rahman.

**Resources:** Md. Siddikur Rahman, Arman Hossain Chowdhury.

**Software:** Md. Siddikur Rahman, Arman Hossain Chowdhury.

**Supervision:** Md. Siddikur Rahman.

**Validation:** Md. Siddikur Rahman, Arman Hossain Chowdhury.

**Visualization:** Md. Siddikur Rahman, Arman Hossain Chowdhury.

**Writing – original draft:** Md. Siddikur Rahman, Arman Hossain Chowdhury.

**Writing – review & editing:** Md. Siddikur Rahman, Arman Hossain Chowdhury.

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
