## [Decision Letter · Decision Letter 0]

24 Jun 2022

PONE-D-22-14088Identification of significant meteorological risk factors and accuracy comparison of ARIMAX and XGBoost models for COVID-19 prediction in SAARC CountriesPLOS ONE

Dear Dr. Rahman,

Thank you for submitting your manuscript to PLOS ONE. After careful consideration, we feel that it has merit but does not fully meet PLOS ONE’s publication criteria as it currently stands. Therefore, we invite you to submit a revised version of the manuscript that addresses the points raised during the review process.

We look forward to receiving your revised manuscript.

Kind regards,

Randeep Singh

Academic Editor

PLOS ONE

Journal Requirements:

2. Please amend either the abstract on the online submission form (via Edit Submission) or the abstract in the manuscript so that they are identical.

3. Please ensure that you refer to Figure 1 in your text as, if accepted, production will need this reference to link the reader to the figure.

4. We note that Figure 2 in your submission contain [map/satellite] images which may be copyrighted. All PLOS content is published under the Creative Commons Attribution License (CC BY 4.0), which means that the manuscript, images, and Supporting Information files will be freely available online, and any third party is permitted to access, download, copy, distribute, and use these materials in any way, even commercially, with proper attribution. For these reasons, we cannot publish previously copyrighted maps or satellite images created using proprietary data, such as Google software (Google Maps, Street View, and Earth). For more information, see our copyright guidelines: http://journals.plos.org/plosone/s/licenses-and-copyright.

Reviewers' comments:

Reviewer's Responses to Questions

**Comments to the Author**

1. Is the manuscript technically sound, and do the data support the conclusions?

Reviewer #1: Partly

Reviewer #2: Partly

Reviewer #3: Yes

Reviewer #4: Yes

2. Has the statistical analysis been performed appropriately and rigorously? 

Reviewer #1: No

Reviewer #2: Yes

Reviewer #3: Yes

Reviewer #4: Yes

3. Have the authors made all data underlying the findings in their manuscript fully available?

Reviewer #1: Yes

Reviewer #2: Yes

Reviewer #3: Yes

Reviewer #4: Yes

4. Is the manuscript presented in an intelligible fashion and written in standard English?

Reviewer #1: No

Reviewer #2: Yes

Reviewer #3: Yes

Reviewer #4: Yes

5. Review Comments to the Author

Reviewer #1: This study found that meteorological factors had both positive and negative impacts on the transmission of COVID-19 confirmed cases. However, to conclude this, the study needs a rigorous regression analysis to control other essential covariates (demographic, socioeconomic, healthcare, human mobility, etc.).

In Table 1, some significant values at the 5% level are inconsistent with the confidence intervals. If a confidence interval contains zero, there is strong evidence that there is no 'significant' difference between the two population means.

There are typographical or grammatical errors, including the agent's name for COVID-19. And finally, the authors need to explain their research findings as the evidence to date is inconsistent.

Reviewer #2: Title is not so attractive and easily understandable. Abbreviation need to be avoided from the title. The tile is large enough, so please concise and make it attractive for laymen reader also.

Abstract- Methodology part need to be more elaborately mentioned, how data was analysis need to be mentioned. The conclusion section is more general rather need to mention about what specific action need to be taken.

Introduction- Section is very short, how climate is related to COVID-19 need to be more emphasized in introduction.

Methodology- Though this is a secondary source of data, but not clearly mentioned. How and from where the permission was taken from the data utilization need to be mentioned. What are the parameters use for the study need to be added. Ethical the ethical permission was taken for this study.

Result- Interesting findings. But why country wise the result is different can be a matter of fact. In Afghanistan and Pakistan the transmission of COVID-19 cases negatively impact with minimum temperature, but reverse in other two countries but why, is there any other factors. Need to measure again the degree of error in your study findings. The findings could not equally achieve all three objective you mentioned.

Conclusion- Is not so attractive, need rewrite ignoring general sentence in such special innovative research. What we can do in future should be directed in the conclusion.

Reviewer #3: The study aimed at identifying significant meteorological risk factors and accuracy comparison of ARIMAX and XGBoost models for COVID-19 prediction in SAARC Countries. The study was well designed. They used sophisticated machine learning algorithms for doing so and from my expertise I think that they have been executed accurately. The results were presented in a well-organized manner. The findings are well-grounded and will be very helpful for the scientific community working the field of COVID-19. Therefore, I am recommending this study for publication.

Reviewer #4: The paper is interesting, however, the key contribution is not clear. Please add additional paragraph of theoretical / practical significance of the research. Also, please copy-edit the entire manuscript. The references are not followed proper guideline:

For example:

175 29. Hyndman RJ. AG. Forecasting: principles and practice. [cited 28 Feb 2022]. Available:

176 https://books.google.com.bd/books?hl=en&lr=&id=_bBhDwAAQBAJ&oi=fnd&pg=PA7

177 &dq=Forecasting+principles+and+practice&ots=Tii-

178 tiZPFJ&sig=Q6Y58Rd860QSTBEi1BxeriFw4Z0&redir_esc=y#v=onepage&q=Forecastin

179 g principles and practice&f=false

180 30. Franses PH. Primary Demand for Beer in The Netherlands : An Application of ARMAX

181 Model. 1991;xxvm: 240–245.

182 31. Lv CX, An SY, Qiao BJ, Wu W. Time series analysis of hemorrhagic fever with renal

183 syndrome in mainland China by using an XGBoost forecasting model. BMC Infect Dis.

184 2021;21: 1–13. doi:10.1186/S12879-021-06503-Y/TABLES/5

6. PLOS authors have the option to publish the peer review history of their article (what does this mean?). If published, this will include your full peer review and any attached files.

Reviewer #1: No

Reviewer #2: **Yes: **Abu Sayeed Md. Abdullah

Reviewer #3: **Yes: **Ishtiaque Ahammad

Reviewer #4: **Yes: **Syed Far Abid Hossain

---

## [Decision Letter · Decision Letter 1]

21 Jul 2022

PONE-D-22-14088R1Time series prediction of COVID-19 transmission in SAARC countries using a data-driven XGBoost machine learning modelPLOS ONE

Dear Dr. Rahman,

Thank you for submitting your manuscript to PLOS ONE. After careful consideration, we feel that it has merit but does not fully meet PLOS ONE’s publication criteria as it currently stands. Therefore, we invite you to submit a revised version of the manuscript that addresses the points raised during the review process.

We look forward to receiving your revised manuscript.

Kind regards,

Randeep Singh

Academic Editor

PLOS ONE

Reviewers' comments:

Reviewer's Responses to Questions

**Comments to the Author**

1. If the authors have adequately addressed your comments raised in a previous round of review and you feel that this manuscript is now acceptable for publication, you may indicate that here to bypass the “Comments to the Author” section, enter your conflict of interest statement in the “Confidential to Editor” section, and submit your "Accept" recommendation.

Reviewer #1: All comments have been addressed

Reviewer #2: All comments have been addressed

Reviewer #4: All comments have been addressed

2. Is the manuscript technically sound, and do the data support the conclusions?

Reviewer #1: No

Reviewer #2: Yes

Reviewer #4: Yes

3. Has the statistical analysis been performed appropriately and rigorously? 

Reviewer #1: No

Reviewer #2: Yes

Reviewer #4: Yes

4. Have the authors made all data underlying the findings in their manuscript fully available?

Reviewer #1: Yes

Reviewer #2: Yes

Reviewer #4: Yes

5. Is the manuscript presented in an intelligible fashion and written in standard English?

Reviewer #1: Yes

Reviewer #2: No

Reviewer #4: Yes

6. Review Comments to the Author

Reviewer #1: The key findings of this study are based on Table 2, which is not valid and not reliable with the description. The authors failed to interpret the findings in Table 2. Most of the beta values are statistically insignificant but described as significant findings in the manuscript. There is a close relationship between confidence intervals and significance tests. Specifically, if a statistic is significantly different from 0 at the 0.05 level, then the 95% confidence interval will not contain 0.

Reviewer #2: Title need to be changed for making more attractive. The abbreviation used in the title need to remove. Three objectives are more in a article which might confuse the reader. The objective need to make easy to understand. How study finings guide the authorities need to mention in conclusion of abstract. The methodology is relatively complex to understand. Conclusion should include some points of recommendation

Reviewer #4: The author (s) tried to investigate the Time series prediction of COVID-19 transmission in SAARC countries using a datadriven XGBoost machine learning model. The revised version is good.

7. PLOS authors have the option to publish the peer review history of their article (what does this mean?). If published, this will include your full peer review and any attached files.

Reviewer #1: **Yes: **Farid Uddin Ahmed

Reviewer #2: No

Reviewer #4: **Yes: **Dr. Syed Far Abid Hossain

---

## [Decision Letter · Decision Letter 2]

8 Aug 2022

A data-driven eXtreme Gradient Boosting machine learning model to predict COVID-19 transmission with meteorological drivers

PONE-D-22-14088R2

Dear Dr. Rahman,

We’re pleased to inform you that your manuscript has been judged scientifically suitable for publication and will be formally accepted for publication once it meets all outstanding technical requirements.

Kind regards,

Randeep Singh

Academic Editor

PLOS ONE

Additional Editor Comments (optional):

Reviewers' comments:

Reviewer's Responses to Questions

**Comments to the Author**

1. If the authors have adequately addressed your comments raised in a previous round of review and you feel that this manuscript is now acceptable for publication, you may indicate that here to bypass the “Comments to the Author” section, enter your conflict of interest statement in the “Confidential to Editor” section, and submit your "Accept" recommendation.

Reviewer #1: All comments have been addressed

Reviewer #2: All comments have been addressed

2. Is the manuscript technically sound, and do the data support the conclusions?

Reviewer #1: Yes

Reviewer #2: Partly

3. Has the statistical analysis been performed appropriately and rigorously? 

Reviewer #1: Yes

Reviewer #2: Yes

4. Have the authors made all data underlying the findings in their manuscript fully available?

Reviewer #1: Yes

Reviewer #2: Yes

5. Is the manuscript presented in an intelligible fashion and written in standard English?

Reviewer #1: Yes

Reviewer #2: No

6. Review Comments to the Author

Reviewer #1: Interestingly and surprisingly the authors have addressed previous issues regarding the analysis. Now it would be better, if they clarify what were the problem-either with the data or with the analysis method!!!!

Reviewer #2: Title can be revised for understanding easily. One main objective can be highlighted in the research. The conclusion of the study is not strong enough. The discussion section need to be elaborated with other such epidemiological distribution of COVID-19 influencing factors. Your highlighted findings are better to justify with other such findings in other countries.

7. PLOS authors have the option to publish the peer review history of their article (what does this mean?). If published, this will include your full peer review and any attached files.

Reviewer #1: **Yes: **Farid Uddin Ahemd

Reviewer #2: No

---

## [Editor Report · Acceptance letter]

2 Sep 2022

PONE-D-22-14088R2 

A data-driven eXtreme Gradient Boosting machine learning model to predict COVID-19 transmission with meteorological drivers 

Dear Dr. Rahman:

I'm pleased to inform you that your manuscript has been deemed suitable for publication in PLOS ONE. Congratulations! Your manuscript is now with our production department. 

Kind regards, 

on behalf of

Dr. Randeep Singh 

Academic Editor

PLOS ONE